# Inferring Networks From Random Walk-Based Node Similarities

**Jeremy G. Hoskins**
Department of Mathematics
Yale University
New Haven, CT
jeremy.hoskins@yale.edu

**Cameron Musco**
Microsoft Research
Cambridge, MA
camusco@microsoft.com

**Christopher Musco**
Department of Computer Science
Princeton University
Princeton, NJ
cmusco@cs.princeton.edu

**Charalampos E. Tsourakakis**
Department of Computer Science
Boston University & Harvard University
Boston, MA
ctsourak@bu.edu

## Abstract

Digital presence in the world of online social media entails significant privacy risks [31, 56]. In this work we consider a privacy threat to a social network in which an attacker has access to a subset of *random walk-based node similarities*, such as effective resistances (i.e., commute times) or personalized PageRank scores. Using these similarities, the attacker seeks to infer as much information as possible about the network, including unknown pairwise node similarities and edges.

For the effective resistance metric, we show that with just a small subset of measurements, one can learn a large fraction of edges in a social network. We also show that it is possible to learn a graph which accurately matches the underlying network on *all other effective resistances.* This second observation is interesting from a data mining perspective, since it can be expensive to compute all effective resistances or other random walk-based similarities. As an alternative, our graphs learned from just a subset of effective resistances can be used as surrogates in a range of applications that use effective resistances to probe graph structure, including for graph clustering, node centrality evaluation, and anomaly detection.

We obtain our results by formalizing the graph learning objective mathematically, using two optimization problems. One formulation is convex and can be solved provably in polynomial time. The other is not, but we solve it efficiently with projected gradient and coordinate descent. We demonstrate the effectiveness of these methods on a number of social networks obtained from Facebook. We also discuss how our methods can be generalized to other random walk-based similarities, such as personalized PageRank scores. Our code is available at https://github.com/cnmusco/graph-similarity-learning.

## 1 Introduction

In graph mining and social network science, a variety of measures are used to quantify the similarity between nodes in a graph, including the shortest path distance, Jaccard's coefficient between node neighborhoods, the Adamic-Adar coefficient [2], and hub-authority-based metrics [30, 9].

An important family of similarity measures are based on random walks, including SimRank [23], random walks with restarts [50], commute times [18], personalized PageRank [39, 24], and DeepWalk embeddings [40]. These measures capture both local and global graph structure and hence are widely used in graph clustering and community detection [4, 44], anomaly detection [42], collaborative filtering [18, 45, 55], link prediction [35], computer vision [20], and many other applications.

In this work we focus on these random walk-based similarity metrics. We initiate the study of a fundamental question:

> How much information about a network can be learned given access to *a subset of potentially noisy estimates* of pairwise node similarities?

This question is important from a privacy perspective. A common privacy breach is *social link disclosure* [6, 56], in which an attacker attempts to learn potentially sensitive links between nodes in a network. Such attacks are very common; fake accounts are used to infiltrate social groups, potential employers may want to inspect a job candidate's social network, and advertisers may wish to probe a user's information to offer targeted ads. Thus, studying the ability of an attacker to reveal link information using node similarities is important in understanding the privacy implications of releasing similarities, or information that can be used to compute them.

There are many scenarios in which node similarities may be released, either directly or indirectly, with the potential to reveal private link information. For example, when searching for users on a social network platform, node similarity is indirectly revealed since similar users (in terms of social connections) are often displayed together in search results. As a second example, random walk-based graph embeddings (e.g. PageRank or DeepWalk embeddings) may be released publicly for research purposes since, naively, they appear to contain no identifying information.

From a data mining perspective, computing all pairwise node similarities can be infeasible for large networks since the number of similarities grows quadratically in the number of nodes. Additionally, when the network cannot be accessed in full but can only be probed via crawling [28], we may only have access to similarity estimates rather than their exact values. Thus, understanding what information can be learned from a partial, potentially noisy, set of node similarities is important when using these metrics in large scale graph mining.

Finally, in some scenarios, it may be possible to measure node similarities for an underlying graph, which we cannot directly access but wish to recover. For example, in evolutionary ecology, effective resistance distances in planar "environment graphs" have been shown to correlate with genetic differentiation in geographically distributed populations [36, 37, 41]. In this context, measurements of geographic genetic variation give incomplete and noisy measurements of effective resistances. Recovering an underlying graph from these measurements corresponds to recovering plausible limitations on migration and movement that could have led to the observed genetic variations.

## 1.1 Learning from Effective Resistances

In this paper, we focus on commute times, which are one of the most widely used random walk-based similarities. Commute times are a scaled version of *effective resistances*, they form a metric, and have many algorithmic applications [47]. Our ideas can be extended to related similarity measures, such as personalized PageRank, which we discuss in Appendix E. It was shown in the seminal work of Liben-Nowell and Kleinberg that effective resistances can be used to predict a significant fraction of future links appearing in networks from existing links, typically ranging from 5% up to 33% [35].

A difficulty associated with this task is that, in contrast to *local* similarity measures such as the number of common neighbors or the Adamic-Adar coefficient [2], node similarity under the effective resistance metric does not necessarily imply local connectivity. For example, two nodes connected by many long paths may be more similar than two nodes directly connected by a single edge.

Furthermore, in certain cases, the effective resistance between two nodes $u, v$ tends to correlate well with a simple function of the degree sequence (specifically, $\frac{1}{d(u)} + \frac{1}{d(v)}$) [52, 53], and it is known that there are many graphs with the same degree sequence but very different global structures. Nevertheless, *considered in aggregate*, effective resistances encode global structure in a very strong way. For any graph, given all pairs effective resistances, it is possible to provably recover the full graph in polynomial time [46, 54]! This contrasts with purely local similarity metrics, which can be used heuristically for link prediction, but do not give network reconstruction in general. For instance, all-pairwise counts of common neighbors in any triangle free graph equal $0$, and thus they reveal no information about graph structure.

While the full information case is well understood, when all exact effective resistances are not available, little is known about what graph information can be learned. Some work considers

reconstruction of trees based on a subset of effective resistances [15, 7, 48]. However outside of this special case, essentially nothing is known.

**Related Work.** Our work is closely related to work on link prediction, graph reconstruction, and phylogenetic tree reconstruction from pairwise distances. We give an overview in Appendix A.

## 1.2 Our Contributions

We study theoretically and empirically what can be learned about a graph given a subset of potentially noisy effective resistance estimates. Our main contributions are:

**Mathematical formulation.** We provide an optimization-based formulation of the problem of learning a graph from effective resistances. Specifically, given a set of effective resistance measurements, we seek a graph whose effective resistances *match the given resistances as closely as possible.*

In general, there may be many different graphs which match any subset of all pairs effective resistances, and hence many minimizers to our optimization problem. If the resistances additionally have some noise, there may be no graph which matches them exactly but many which match them approximately. Nevertheless, as we show empirically, the graph obtained via our optimization approach typically recovers significant information about the underlying graph, including a large fraction of its edges, its global structure, and good approximations to *all* of its effective resistances.

**Algorithms.** We prove that, in some cases, the optimization problem we present can be solved exactly, in polynomial time. However, in general, the problem is non-convex and does not admit an obvious polynomial time solution. We show that it can be solved via iterative methods along with a powerful initialization strategy that allows us to find high quality solutions in most instances.

We also show that the problem can be relaxed to a convex formulation. Instead of searching for a graph that matches all given effective resistance measurements, we just find a graph whose effective resistances are upper bounded by those given and which has minimum total edge weight. This modified problem is convex and can be solved via an SDP.

**Experimental Results.** We evaluate our algorithms on synthetic graphs and real Facebook ego networks, which contain all nodes in the social circle of a user. Ego networks are important in many applications and allow us to effectively test our ability to recover local graph structure. We show that, given a small randomly selected fraction of all effective resistance pairs (10%-25%), we can learn a large fraction of a network – typically between 20% and 60% of edges, even after adding noise to the given effective resistances.

We also show that by finding a graph which closely matches the given set of effective resistances (via our optimization approach), we in fact find a graph which closely matches the underlying network on *all* effective resistance pairs. This indicates that significant information contained in all pairs effective resistances can be learned from just a small subset of these pairs, even when corrupted by noise.

## 2 Proposed Method

### 2.1 Notation and Preliminaries

For an undirected, weighted graph $G = (V, E, w)$ with $n$ nodes, we let $A$ be the $n \times n$ adjacency matrix. $L$ denotes the graph Laplacian: $L = D - A$, where $D$ is a diagonal matrix with $D_{i,i}$ equal to the weighted degree of node $i$. For an integer $n > 0$, $[n]$ denotes the set $\{1, 2, ..., n\}$. $e_i$ denotes the $i^{th}$ standard basis vector. For a matrix $M$, $M_{i,j}$ denotes the entry in its $i^{th}$ row and $j^{th}$ column.

**Commute time and effective resistance.** For two nodes $u, v \in V$, the hitting time $h_G(u, v)$ is the expected time it takes a random walk to travel from $u$ to $v$. The *commute time* is its symmetrized version $c_G(u, v) = h_G(u, v) + h_G(v, u)$, i.e., the time to move from $u$ to $v$ and then back to $u$. For connected graphs, the effective resistance between $u, v$ is a scaling of the commute time: $r_G(u, v) = \frac{c_G(u,v)}{\text{vol}(G)}$ where $\text{vol}(G) = 2 \sum_{e \in E} w_e$. Effective resistance has a natural electrical interpretation. When $G$ is viewed as an electrical network on $n$ nodes where each edge $e$ corresponds to a link of conductance $w_e$ (equivalently to a resistor of resistance $\frac{1}{w_e}$), the effective resistance is the

voltage difference that appears across $u, v$ when a unit current source is applied to them. Effective resistances (and hence commute times) always form a metric [29].

Let $\chi_{u,v} = e_u - e_v$. The effective resistance between nodes $u$ and $v$ in a graph $G$ with Laplacian $L$ is

$$r_G(u, v) = \chi_{u,v}^T L^+ \chi_{u,v}. \tag{1}$$

Here $L^+$ denotes the Moore-Penrose pseudoinverse of $L$.

## 2.2 Problem Definition

We begin by providing a mathematical formulation of the problem introduced in Section 1 – that of learning the structure of a graph from partial and possibly noisy measurements of pairwise effective resistances. An analogous problem can be defined for other random walk-based similarities, such as personalized PageRank. We discuss initial results in this direction in supplementary Appendix E.

> **Problem 1** (Graph Reconstruction From Effective Resistances). *Reconstruct an unknown graph $G$ given a set of noisy effective resistance measurements,*
>
> $$\bar{r}(u, v) = r_G(u, v) + n_{uv}$$
>
> *for each $(u, v) \in \mathcal{S}$, where $\mathcal{S} \subseteq [n] \times [n]$ is a set of node pairs and $n_{uv}$ is a random noise term.*

We focus on three interesting cases of Problem 1:

**Problem 1.1** $\mathcal{S} = [n] \times [n]$ and $n_{uv} = 0$ for all $(u, v) \in S$. This is the *full information setting.*

**Problem 1.2** $\mathcal{S}$ is a subset of $[n] \times [n]$ and $n_{uv} = 0$ for all $(u, v) \in \mathcal{S}$. In this setting we must learn $G$ from a limited number of exact effective resistances.

**Problem 1.3** $\mathcal{S}$ is a subset of $[n] \times [n]$ and $n_{uv}$ is a random term, e.g. a mean $0$ normal random variable with variance $\sigma^2$: $n_{uv} \sim \mathcal{N}(0, \sigma^2)$.

It is known that there in a unique graph consistent with any full set of effective resistance measurements (see e.g., [46, 54]). Additionally, this graph can be computed by solving a fully determined linear system. So, we can solve Problem 1.1 exactly in polynomial time (see Section 3.1).

From a privacy and data mining perspective, the limited information settings of Problems 1.2 and 1.3 are more interesting. In Section 3.1 we show that, when $G$ is a tree, exact recovery is possible for Problem 1.2 when $\mathcal{S}$ is a superset of $G$'s edges. However, in general, there is no closed form solution to these problems, and exact recovery of $G$ is typically impossible – several graphs may be consistent with the measurements given. We address these cases by reposing Problem 1 as an optimization problem, in which we attempt to recover a graph matching the given resistances as best as possible.

## 2.3 Optimization Formulation

A natural formalization of Problem 1 is as a least squares problem.

> **Problem 2.** *Given a set of vertex pairs $\mathcal{S} \subseteq [n] \times [n]$ and a target effective resistance $\bar{r}(u, v)$ for each $(u, v) \in \mathcal{S}$:*
>
> $$\underset{\text{graph } H}{\text{minimize}} \quad F(H) \overset{\text{def}}{=} \sum_{(u,v) \in \mathcal{S}} \left[ r_H(u, v) - \bar{r}(u, v) \right]^2. \tag{2}$$

Using formula (1) for effective resistances, Problem 2 can equivalently be viewed as an optimization problem over the set of graph Laplacians: minimize $\sum_{(u,v) \in \mathcal{S}} \left[ \chi_{u,v}^T L^+ \chi_{u,v} - \bar{r}(u, v) \right]^2$. While this set is convex, the objective function is not and it is unclear if it can be minimized provably in polynomial time. Nevertheless, we show in Section 3.2 that it is possible to solve the problem approximately by combining projected gradient and coordinate descent algorithms with a powerful initialization heuristic. This approach quickly converges to near global minimums for many networks.

For Problem 1.2, where $\bar{r}(u, v)$ comprise a subset of the exact effective resistances for some graph $G$, $\min_H F(H) = 0$. This minimum may be achieved by multiple graphs (including $G$) if $\mathcal{S}$ does not

contain all effective resistance pairs. Nevertheless, we show experimentally in Section 4 that even when $\mathcal{S}$ contains a small fraction of these pairs, an approximate solution to Problem 2 often recovers significant information about $G$, including a large fraction of its edges. Interestingly, while Problem 2 only minimizes over the subset $\mathcal{S}$, the recovered graph typically matches $G$ on *all* effective resistances, explaining why it contains so much structural information. For Problem 1.3, if $\mathcal{S} = [n] \times [n]$ and the noise terms $n_{uv}$ are distributed as i.i.d. Gaussians, Problem 2 gives the maximum likelihood estimator for $G$. We again show that an approximate solution can recover many of $G$'s edges.

We note that while we can solve Problem 2 quickly via iterative methods, we leave open provable polynomial time algorithms in the settings of both Problems 1.2 and 1.3.

**Convex relaxation.** As an alternative to Problem 2, we give an optimization formulation of Problem 1 that *is convex*. Here we optimize over the convex set of graph Laplacians.

---

**Problem 3.** *Let $\mathcal{L}$ be the convex set of $n \times n$ graph Laplacians. Given a set of vertex pairs $\mathcal{S} \subseteq [n] \times [n]$ and a target effective resistance $\bar{r}_{(u,v)}$ for every $(u, v) \in \mathcal{S}$,*

$$\underset{L \in \mathcal{L}}{\text{minimize}} \qquad \text{Tr}(L)$$

$$\text{subject to} \qquad \chi_{u,v}^T L^+ \chi_{u,v} \leq \bar{r}(u, v) \quad \forall (u, v) \in \mathcal{S}$$

---

By Rayleigh's monotonicity law, decreasing the weight on edges in $L$ increases effective resistances. $\text{Tr}(L)$ is equal to the total degree of the graph corresponding to $L$, so the problem asks us to find a graph with as little total edge weight as possible that still satisfies the effective resistance constraints.

The disadvantage of Problem 3 is that it only encodes the target resistances $\bar{r}(u, v)$ as *upper bounds* on the resistances of $L$. The advantage is that we can solve it provably in polynomial time via semidefinite programming (see supplemental Appendix C). In practice, we find that it can sometimes effectively learn graph edges and structure from limited measurements.

Problem 3 is related to work on convex methods for minimizing total effective resistance or relatedly, mixing time in graphs [10, 49, 19]. However, prior work does not consider pairwise resistance constraints and so is not suited to graph learning.

## 3 Analytical Results and Algorithms

### 3.1 Full Graph Reconstruction – Problem 1

Problem 1 can be solved exactly in polynomial time when $\mathcal{S}$ contains *all resistance pairs* of some graph $G$ (i.e. Problem 1.1). In this case, there is a closed form solution for $G$'s Laplacian $L$ and the solution is unique. This was pointed out in [46], however we include our own proof in the supplementary Appendix B for completeness.

**Theorem 1** (Lemma 9.4.1. of [46])**.** *If there is a feasible solution to Problem 1.1, it is unique and can be found in $O(n^3)$ time. Let $R$ be a matrix with $R_{u,v} = r_G(u, v)$ for all $u, v \in [n]$. The Laplacian $L$ of the solution $G$ is*

$$-2 \cdot \left[ \left( I - \frac{J}{n} \right) R \left( I - \frac{J}{n} \right) \right]^+. \tag{3}$$

*Here $I$ is the $n \times n$ identity matrix and $J$ is the $n \times n$ all ones matrix.*

**Reconstruction from hitting times.** The above immediately generalizes to graph reconstruction from hitting times since, as discussed, for connected $G$, the effective resistance between $u, v$ can be written as $r_G(u, v) = \frac{c_G(u,v)}{vol(G)} = \frac{h_G(u,v) + h_G(v,u)}{vol(G)}$. Thus, by Theorem 1, we can recover $G$ up to a scaling from all pairs hitting times. This recovers a result in [54].

**Reconstruction from other similarity measures.** An analogous result to Theorem 1 holds for graph recovery from all pairs personalized PageRank scores, and for related measures such as Katz similarity scores [27]. We discuss this direction in supplementary Appendix E.

**Are all pairs always necessary for perfect reconstruction?** For general graphs, Problem 1 can only be solved exactly when $\mathcal{S}$ contains all $\binom{n}{2}$ true effective resistances. However, given additional

constraints on $G$, recovery is possible with much less information. In particular, we show in Appendix B that when $G$ is a tree, we can recover it (i.e., solve Problem 1.2) if $\mathcal{S}$ is a superset of its edge set. The problem of recovering trees from pairwise distances is a central problem in phylogenetics.

### 3.2   Graph Learning via Least Squares Minimization – Problem 2

When Problem 1 cannot be solved exactly, e.g. in the settings of Problems 1.2 and 1.3, an effective surrogate is to solve Problem 2 to find a graph with effective resistances close to the given target resistances. As we demonstrate experimentally in Section 4, this yields good solutions to Problems 1.2 and 1.3 in many cases. Problem 2 is non-convex, however we show that a good solution can often be found efficiently via projected gradient descent.

**Optimizing over edge weights.** Let $m = \binom{n}{2}$. We write the Laplacian of the graph $H$ as $L(w) \stackrel{\text{def}}{=} B^T \operatorname{diag}(w) B$, where $w \in \mathbb{R}^m$ is a non-negative vector whose entries correspond to the edge weights in $H$, $\operatorname{diag}(w)$ is the $m \times m$ matrix with $w$ as its diagonal, and $B \in \mathbb{R}^{m \times n}$ is the vertex edge incidence matrix with a row equal to $\chi_{u,v} = e_u - e_v$ for every possible edge $(u, v) \in [n] \times [n]$.

Optimizing the objective function $F(H)$ in Problem 2 is equivalent to optimizing $F(w)$ over the edge weight vector $w$, where we define $F(w) = F(H)$ for the unique $H$ with Laplacian equal to $L(w)$.

We restrict $w_i \geq 0$ for all $i$ and project to this constraint after each gradient step by setting $w_i := \max(w_i, 0)$. The gradient of $F(w)$ can be computed in closed form. We first define the variable, $R(w) \in \mathbb{R}^{m \times m}$, whose diagonal contains all effective resistances of $H$ with weight vector $w$:

**Definition 1.** *For $w \in \mathbb{R}^m$ with $w_i \geq 0$ for all $i$, define $R(w) = BL(w)^+ B^T$.*

Using $R(w)$ we can compute the gradient of $F(w)$ by:

**Proposition 1.** *Let $\circ$ denote the Hadamard (entrywise) product for matrices. Define the error vector $\Delta(w) \in \mathbb{R}^m$ as having $\Delta(w)_i = \bar{r}(i) - [R(w)]_{i,i}$ for all $i \in \mathcal{S}$ and 0s elsewhere. We have:*

$$\nabla F(w) = 2 \left( R(w) \circ R(w) \right) \Delta(w).$$

We give a proof in Appendix B, along with a formula for the Hessian of $F(w)$.

**Acceleration via coordinate descent.** Naively computing the gradient $\nabla F(w)$ via Proposition 1 requires computing the full $m \times m$ matrix $R(w)$, which can be prohibitively expensive for large graphs – recall that $m = \binom{n}{2} = O(n^2)$. Note however, that the error vector $\Delta(w)$ only has nonzero entries at positions corresponding to the node pairs in $\mathcal{S}$. Thus, it suffices to compute just $|\mathcal{S}|$ columns of $R(w)$ corresponding to these pairs, which can give a significant savings. We obtain further savings using block coordinate descent. At each step we restrict our updates to a random subset of edges $\mathcal{B} \subseteq [n] \times [n]$, and so only form the rows of $R(w)$ corresponding to these edges.

**Initialization.** A good initialization can significantly accelerate the solution of Problem 2. We use a strategy based on the exact solution to Problem 1.1 in Theorem 1.

Since effective resistances form a metric, by triangle inequality, for any $u, v, w \in [n]$, $r_H(u, v) \leq r_H(u, w) + r_H(w, v)$. Guided by this fact, given targets $\bar{r}(u, v)$ for $(u, v) \in \mathcal{S}$, we first "fill in" the constraint set. For $(w, z) \notin \mathcal{S}$, we set $\bar{r}(w, z)$ equal to the shortest path distance in the graph $\bar{G}$ which has an edge for each pair in $\mathcal{S}$ with length $\bar{r}(u, v)$.

We thus obtain a full set of target effective resistances. We can form $R$ with $R_{u,v} = \bar{r}(u, v)$ and initialize the Laplacian of $H$ using the formula given in (3) in Theorem 1. However, this formula is quite unstable and generally yields an output which is far from a Laplacian even when $R$ is corrupted by a small amount of noise. So we instead compute for some $\lambda > 0$, a regularized estimate, $\tilde{L} = -2 \cdot \left[ \left( I - \frac{J}{n} \right) R \left( I - \frac{J}{n} \right) + \lambda I \right]^+$. Generally, $\tilde{L}$ will not be a valid graph Laplacian, so we remove any negative edge weights to obtain our initialization.

## 4   Empirical results

We next present an experimental study of how well our methods can learn a graph given a set of (noisy) effective resistance measurements. We focus on two key questions:

1. Given a set of effective resistance measurements, can we find a graph matching these measurements via the optimization formulations of Problems 2 and 3 and the algorithms of Section 3.2?

2. What structure does the graph we learn share with the underlying network that produced the resistance measurements? Can it be used to uncover links? Does it approximately match the network on effective resistances outside the measurement set, or share other global structure?

## 4.1 Experimental Setup

We address these questions by examining a variety of graphs. We study two synthetic examples: an $8 \times 8$ grid graph and a $k$-nearest neighbor graph constructed for vectors drawn from a Gaussian mixture model with two clusters. We also consider Facebook 'ego networks' obtained from the Stanford Network Analysis Project (SNAP) collection [33, 34]. Each of these networks is formed by taking the largest connected component in the social circle of a specific user. Details on the networks studied are shown in Table 2 in supplemental Appendix D.

For all experiments, we provide our algorithms with effective resistances uniformly sampled from the set of all $\binom{n}{2}$ effective resistances. We sample a fixed fraction $f \stackrel{\text{def}}{=} \left(|\mathcal{S}|/\binom{n}{2}\right) \times 100\%$ of all possible measurements. We typically use $f \in \{10, 25, 50, 100\}\%$. In some cases, these resistances are corrupted with i.i.d. Gaussian noise $\eta \sim \mathcal{N}(0, \sigma^2)$. We experiment with different values of $\sigma^2$.

For Problem 2 we implemented gradient decent based on the closed-from gradient calculation in Proposition 1. Line search was used to optimize step size at each iteration. For larger problems, block coordinate descent was used as described in Section 3.2, with the coordinate set chosen uniformly at random in each iteration. We set the block size $|\mathcal{B}| = 5000$. For Problem 3 we used the MOSEK convex optimization software, accessed through the CVX interface [38, 21]. All experiments were run on a laptop with a 2.6 GHz Intel Core i7 processor and 16 GB of main memory.

## 4.2 Learning Synthetic Graphs

We first evaluate our algorithms on GRID and K-NN, which are simple graphs with clear structure.

**Least squares formulation.** We first observe that gradient descent effectively minimizes the objective function of Problem 2 on the GRID and K-NN graphs. We consider the normalized objective for constraints $\mathcal{S}$ and output graph $H$:

$$\widehat{F}(H) = \frac{\sum_{(u,v) \in \mathcal{S}} [r_H(u,v) - \bar{r}(u,v)]^2}{\sum_{(u,v) \in \mathcal{S}} \bar{r}(u,v)^2}. \tag{4}$$

For noise variance 0, $\min_H \widehat{F}(H) = 0$ and in Figure 2 we see that for GRID we in fact find $H$ with $\widehat{F}(H) \approx 0$ for varying sizes of $\mathcal{S}$. Convergence is faster when $100\%$ of effective resistances are included in $\mathcal{S}$, but otherwise does not correlate strongly with the number of constraints.

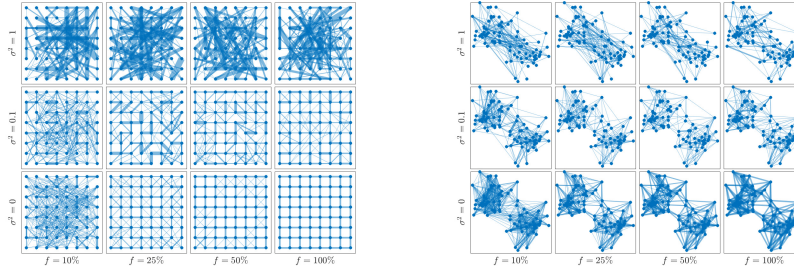

Figure 1: Graphs learned by solving Problem 2 with gradient descent for uniformly sampled effective resistances with varying levels of Gaussian noise. Edge width in plots is proportional to edge weight.

In Figure 2 we also plot the *generalization error*:

$$F_{gen}(H) = \frac{\sum_{(u,v) \in [n] \times [n]} [r_H(u,v) - r_G(u,v)]^2}{\sum_{(u,v) \in [n] \times [n]} r_G^2(u,v)}, \tag{5}$$

where $r_G(u, v)$ is the true effective resistance, uncorrupted by noise. $F_{gen}(H)$ measures how well the graph obtained by solving Problem 2 matches *all* effective resistances of the original network. We confirm that generalization decreases with improved objective function performance, indicating that optimizing Problem 2 effectively extracts network structure from a small set of effective resistances. We observe that the generalization error is small even when $f = 10\%$, and becomes negligible as we increase the fraction $f$ of measurements, even in the presence of noise.

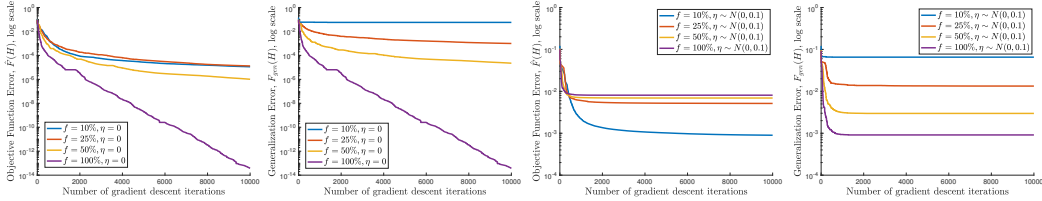

Figure 2: Objective and generalization error for Problem 2 – see (5). For details, see Section 4.2.

We repeat the same experiments with Gaussian noise added to each resistance measurement. The variance of the noise, $\sigma^2$, is scaled relatively to the mean effective resistance in the graph, i.e., we set $\bar{r}(u, v) = r_G(u, v) + \mathcal{N}(0, \bar{\sigma}^2)$ where: $\bar{\sigma}^2 = \frac{\sigma^2}{\binom{n}{2}} \cdot \sum_{(u,v) \in [n] \times [n]} r_G(u, v)$. While generally $\min_H \widehat{F}(H) > 0$ when $\bar{r}(u, v)$ is noisy (since there is no graph consistent with these noisy measurements), the objective value and generalization error still decrease steadily with more iterations.

We obtain similar results by applying Problem 2 to the K-NN graph (see Figure 4 in supplemental Appendix D). Again, gradient descent converges for a variety of noise levels and constraint sets. Convergence leads to improved generalization error.

Figure 1 shows the graphs obtained from solving the problem for varying $\sigma^2$ and $f$. For both graphs, when $\sigma^2 = 0$ and $f = 100\%$, the original network is recovered exactly. Reconstruction accuracy decreases with increasing noise and a decreasing number of constraints. For GRID, even with 25% of constraints, nearly full recovery is possible for $\sigma^2 = 0$ and recovery of approximately half of true edges is possible for $\sigma^2 = 0.1$. For K-NN, for $\sigma^2 = 0$ and $\sigma^2 = 0.1$ we observe that cluster structure is recovered. Detailed quantitative results for both networks are given in Table 1.

**Convex formulation.** We next evaluate the performance of the convex Problem 3. In this case, we do not focus on convergence as we solve the problem directly using semidefinite programming. Unlike for Problem 2, solving Problem 3 does not recover the exact input graph, even in the noiseless all pairs effective resistance case. This is because the input graph does not necessarily minimize the objective – there can be graphs with smaller total edge weight and lower effective resistances.

However, the learned graphs *do capture* information about edges in the original: their heaviest edges typically align with true edges in the target graph. We show quantitative results in Table 1 and qualitative results for GRID in Figure 3. We mark the 224 heaviest edges in the learned graph in red and see that this set converges exactly on the grid.

The convex formulation never significantly outperforms the least squares formulation of Problem 2, and often significantly underperforms. However, we believe there is further opportunity for exploring Problem 3, especially given its provable runtime guarantees.

### 4.3 Learning Social Network Graphs

We conclude by demonstrating the effectiveness of the least squares formulation of Problem 2 in learning Facebook ego networks from limited effective resistance measurements. We consider three metrics of performance, shown in Table 1 for a number of networks learned from randomly sampled subsets of effective resistances, corrupted with noise.

**Objective Function Value:** the value of the normalized objective function (4) of Problem 2.

**Generalization Error:** the error in reconstructing all effective resistances of the true graph (5).

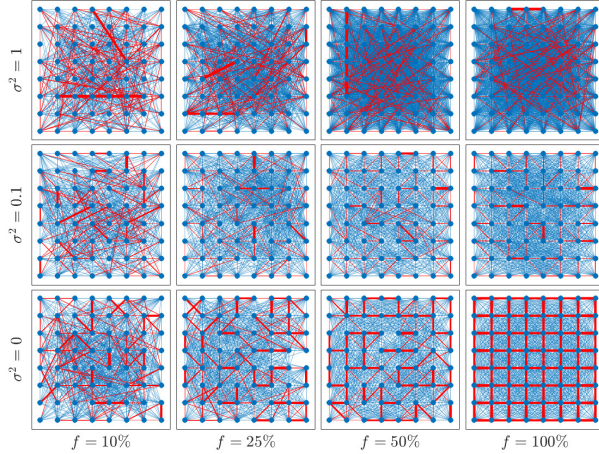

Figure 3: Graphs learned from solving the convex program in Problem 3 for uniformly sampled effective resistances from GRID with varying $f, \sigma^2$. Heaviest edges marked in red.

**Edges Learned:** the rate of recovery for edges in the true graph. We utilize a standard metric from the link prediction literature [30]: given underlying graph $G$ with $m$ edges and learned graph $H$, we consider the $m$ heaviest edges of $H$ and compute the percentage of $G$'s edges contained in this set.

**Results.** We find that as for the synthetic GRID and K-NN graphs, we can effectively minimize the objective function of Problem 2 for the Facebook ego networks. Moreover, this minimization leads to very good generalization error in nearly all cases. i.e., we can effectively learn a graph matching our input on all effective resistances, even when we consider just a small subset.

For all graphs, we are able to recover a significant fraction of edges (typically over $20\%$ ), even when just considering $10\%$ or $25\%$ of effective resistance pairs. We obtain the best recovery for small graphs, learning over half of the true edges in FB SMALL A and FB SMALL C.

Typically, the number of edges learned increases as we increase the number of constraints and decrease the noise variance. However, occasionally, considering fewer effective resistances in fact improves learning, possibly because we more effectively solve the underlying optimization problem.

Table 1: Graph recovery results. All results use a randomly sampled subset of $f = 10\%$ or $25\%$ of all effective resistances. For "Algorithm", GD denotes gradient descent. CD denotes block coordinate descent with random batches of size 5000. "Noise level, $\sigma^2$" indicates that the target resistances were set to $\bar{r}(u,v) = r_G(u,v) + \mathcal{N}(0, \sigma^2 mean_{u,v}(r_G(u,v)))$. "% Edges baseline", is the edge density of the network, equivalent to the expected edge prediction accuracy achieved with random guessing.

| Network | Algorithm | $\sigma^2$ | Objective function error | | Effect. resistance generalization error | | % Edges learned baseline | % Edges learned | |
|---|---|---|---|---|---|---|---|---|---|
| | | | $f = 10\%$ | $f = 25\%$ | $f = 10\%$ | $f = 25\%$ | | $f = 10\%$ | $f = 25\%$ |
| GRID | GD | 0 | .00001 | .00001 | .06559 | .00099 | 5.56 | 20.54 | 88.39 |
| | GD | .1 | .00090 | .00514 | .08129 | .01336 | | 25.89 | 50.00 |
| | SDP | 0 | na | na | .08758 | .07422 | | 16.07 | 25.00 |
| | SDP | .1 | na | na | .09549 | .09343 | | 12.50 | 26.79 |
| K-NN | GD | 0 | .00001 | .00002 | .01122 | .00117 | 11.58 | 44.54 | 72.68 |
| | GD | .1 | .00197 | .00447 | .05536 | .00709 | | 25.96 | 41.53 |
| | SDP | 0 | na | na | .09314 | .10399 | | 27.05 | 48.36 |
| | SDP | .1 | na | na | .11899 | .14097 | | 24.32 | 39.89 |
| FB SMALL A | GD | 0 | .01345 | .00001 | .21097 | .00984 | 28.20 | 44.54 | 75.00 |
| | GD | .1 | .00017 | .00204 | .07964 | .01687 | | 53.64 | 60.00 |
| FB SMALL B | GD | 0 | .00002 | .00003 | .01515 | .00623 | 14.59 | 42.75 | 48.55 |
| | GD | .1 | .00032 | .00206 | .02229 | .01291 | | 36.23 | 43.48 |
| FB SMALL C | GD | 0 | .00162 | .00166 | .00217 | .00203 | 15.55 | 57.03 | 59.51 |
| | GD | .1 | .00532 | .00644 | .01542 | .00218 | | 52.66 | 57.51 |
| FB SMALL D | GD | 0 | .00335 | .00434 | .00821 | .00830 | 11.80 | 21.92 | 24.52 |
| | GD | .1 | .00610 | .18384 | .00923 | .21426 | | 21.38 | 21.20 |
| FB MEDIUM A | GD | 0 | .00447 | .00665 | .02910 | .01713 | 12.78 | 23.50 | 25.59 |
| FB MEDIUM B | CD | 0 | .00224 | .01255 | .00862 | .01471 | 4.80 | 18.97 | 22.15 |
| | CD | .1 | .01174 | .03182 | .01687 | .03295 | | 17.50 | 16.03 |
| FB LARGE A | CD | 0 | .00516 | .00796 | .00682 | .00862 | 3.41 | 10.52 | 12.45 |
| FB LARGE B | CD | 0 | .00524 | .00440 | .00635 | .00580 | 9.51 | 20.26 | 24.95 |
| | CD | .1 | .12745 | .34646 | .14532 | .36095 | | 19.43 | 16.97 |

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
