[Supplementary Material]

# A    Related Work

In this section we give an in depth overview of related work.

**Link prediction and privacy in social networks.** The link prediction problem was popularized by Liben-Nowell and Kleinberg [35]. The goal of *link prediction* is to infer which edges are likely to appear in the near-future, given a snapshot of the network. In [35] various node similarity measures are used to predict a non-trivial fraction of future interactions. Other works focus on predicting positive or negative links [32, 51]. For an extensive survey, see [3].

While related, link prediction differs from the problem we consider since, typically, one is given full access to a network snapshot (and e.g., could compute exact pairwise node similarities for this network) and seeks to make predictions about *future evolutions of the network*. In our setting, we are given partial information about a network (via a partial set of noisy similarities) and seek to learn existing links.

While link prediction is useful in applications ranging from understanding network evolution, to link recommendation, and predicting interactions between terrorists, it entails privacy risks. For instance, a malicious attacker can use link prediction to disclose information about sensitive connections in a sexual contact graph [31, 56]. Private link prediction is not adequately explored in the literature. Abebe and Nakos suggest a possible formalization [1].

**Learning graphs.** Learning graphs from data is central to many disciplines including machine learning [25], network tomography [11], bioinformatics [16], and phylogenetics [17]. The general framework is that there exists a hidden graph that we wish to discover by exploiting some kind of data, e.g., answers from a blackbox oracle returning certain (possibly noisy) graph measurements.

Theoretical work in this area has focused on worst case *query* complexity. Two representative examples include Angluin and Chen's work on learning graphs using edge queries [5], and the recent work of Kannan, Mathieu, and Zhou using distance queries [26]. A number of works also consider learning graph parameters such as node count and mixing time by examining random walks traces [28, 14, 8]. Experimental work has focused on recovering graphs from noisy measurements such as GPS traces [13], and distances between cell populations based on genetic differences [17, 16].

**Learning trees.** While a tree is special type of a graph, the case of learning trees deserves special mention since significant work on learning graphs from data has focused on trees. Distance-based reconstruction of trees aims to reconstruct a phylogenetic tree whose leaves correspond to $n$ species, given their $\binom{n}{2}$ pairwise distances [17]. Note that on a tree, pairwise distances are identical to pairwise effective resistanes. Batagelj et al. study tree realizability assuming access to fewer than $\binom{n}{2}$ leaves' pairwise distances [7]. A spectral method has been proposed by Stone and Griffing [48]. Culberson and Rudnicki [15] consider the problem of reconstructing a degree restricted tree given its distance matrix, see also [43].

# B    Omitted Proofs

**Exact Graph Recovery.** We now give proofs of the exact recovery results of Section 3.1 as well as the gradient and Hessian computations for Problem 2 discussed in Section 3.2.

**Theorem 1.** *If there is a feasible solution to Problem 1.1 then it is unique and can be found in $O(n^3)$ time. Specifically, the Laplacian $L$ of the solution $G$ is given by*

$$-2 \cdot \left[ \left( I - \frac{J}{n} \right) R \left( I - \frac{J}{n} \right) \right]^{+} \tag{6}$$

*where $R$ is the matrix with $R_{u,v} = r_G(u, v)$ for all $u, v \in [n]$, $I$ is the $n \times n$ identity matrix , and $J$ is the $n \times n$ all ones matrix.*

*Proof.* For an $n$ node graph, the number of possible edges is $m = \binom{n}{2}$. Let $B \in \mathbb{R}^{m \times n}$ be the vertex edge incidence matrix of the complete graph with a row equal to $\chi_{u,v} = e_u - e_v$ for every $(u, v)$.

The Laplacian of any $n$ vertex graph can be written as $L = B^T W B$, for some $W \in \mathbb{R}^{m \times m}$ which is a nonnegative diagonal matrix with entries corresponding to the edge weights.

We can rewrite the effective resistance formula in (1) as:

$$r_L(u,v) = \chi_{u,v}^\top L^+ \chi_{u,v} = (L^+)_{u,u} + (L^+)_{v,v} - 2(L^+)_{u,v}. \tag{7}$$

Since $L^+$ is symmetric we need only determine $\frac{n(n+1)}{2}$ unknown entries to determine the full matrix. Moreover, since the all ones vector is in the null space of $L$ and therefore $L^+$, we see that:

$$L_{u,u}^+ = -\sum_{v \neq u} L_{u,v}^+, \tag{8}$$

and hence we can rewrite (7) as:

$$r_{u,v} = -\sum_{v' \neq v, u} (L^+)_{u,v'} - \sum_{u' \neq u,v} (L^+)_{u',v} - 4(L^+)_{u,v}. \tag{9}$$

Let $M$ be the $m \times m$ matrix with rows and columns indexed by pairs $u, v \in [n]$ with $u \neq v$ and $(u_1, v_1), (u_2, v_2)$ entry given by:

$$M_{(u_1,v_1),(u_2,v_2)} = \begin{cases} -4 \text{ if } u_1 = u_2 \text{ and } v_1 = v_2 \\ -1 \text{ if } u_1 = u_2 \text{ or } v_1 = v_2 \\ 0 \text{ otherwise.} \end{cases}$$

Let $r \in \mathbb{R}^m$ contain each effective resistance $r_G(u,v)$. We can see from (9) that if we solve the linear system $Mx = -r$, as long as $M$ is full rank and so the solution is unique, the entries of $x$ will give us each $(L^+)_{u,v}$ with $u \neq v$. We can then use these entries to recover the remaining diagonal entries of $L^+$ using (8).

We can verify that $M$ is in fact always full rank by writing $M = -|B||B|^T - 2I$, where $|B|$ denotes the matrix formed from $B$ by taking the absolute value of each of its entries. We note that the non-zero eigenvalues of $|B||B|^T$ are equal to the non-zero eigenvalues of $|B^T||B|$ which is the $n \times n$ matrix $\overline{M} = (n-2)I + J$. $\overline{M}$ has eigenvalues $2n-2$ with multiplicity 1 and $n-2$ with multiplicity $n-1$. The remaining $m-n$ eigenvalues of $|B||B|^T$ are zero. Consequently, the eigenvalues of $M$ are $-2n$ with multiplicity 1, $-n$ with multiplicity $n-1$ and $-2$ with multiplicity $m-n$. Thus $M$ is full rank, proving that the effective resistances fully determine $L^+$ and thus $L$.

Solving for $L$ via the linear system $Mx = -r$ would require $O(n^6)$ time, however, the closed form solution (6) given in Lemma 9.4.1 of [46] allows us to solve this problem in $O(n^3)$ time. $\qquad \square$

We next show that exact recovery is possible if $G$ is a tree and $\mathcal{S}$ is a superset of $G$'s edge set. Roughly, since $G$ is a tree, the effective resistance $r_G(u,v)$ is equal to the length of the unique path connecting $u$ and $v$. As long as $\mathcal{S}$ includes all edges in $G$, it fully determines all path lengths and hence the effective resistances for all pairs $u, v$. We can thus recover $G$ via Theorem 1.

**Theorem 2.** *If $G$ is a tree and a feasible solution to Problem 1.2 with edge set $E \subseteq \mathcal{S}$ then $G$ is unique and can be found in $O(n^3)$ time.*

*Proof.* Let $P_{uv}$ be the unique path between $u, v$ in $G$. It is well known [12] that:

$$r_G(u,v) = \sum_{e \in P_{uv}} 1/w_e. \tag{10}$$

For $(u,v) \in \mathcal{S}$ set $\bar{r}(u,v) = r_G(u,v)$. Let $\overline{G}$ be an undirected graph with an edge for each $(u,v) \in \mathcal{S}$ with length $r_G(u,v)$. For all $(u,v) \notin \mathcal{S}$, set $\bar{r}(u,v)$ to the shortest path distance between $u$ and $v$ in $\overline{G}$.

**Claim.** $\bar{r}(u,v) = r_G(u,v)$ for all $(u,v) \in [n] \times [n]$.
For *any pair* $(u,v)$, we have $\bar{r}(u,v) \leq r_G(u,v)$. The length of shortest path between $u, v$ in $\overline{G}$ is certainly at most the length of $P_{uv}$, which is contained in $\overline{G}$ since $E \subseteq \mathcal{S}$. $P_{uv}$'s length in $\overline{G}$ is: $\sum_{e \in P_{uv}} r_G(e) = \sum_{e \in P_{uv}} 1/w_e$ using (10). Thus, again via (10),

$$\bar{r}(u,v) \leq \sum_{e \in P_{uv}} 1/w_e = r_G(u,v).$$

Further, $P_{uv}$ is in fact a shortest path between $u, v$ in $\overline{G}$, giving that $\overline{r}(u, v) = r_G(u, v)$. This is because the length of every edge $(u, v) \in \mathcal{S}$ that is not in $E$ just equals the length of path $P_{uv}$ in $\overline{G}$ (i.e., $r_G(u, v)$) and so removing this edge from $\overline{G}$ does not change any shortest path distance. So we can assume that $\overline{G}$ just contains the edges in $E$, and so $P_{uv}$ is the unique path between $u, v$.

Given the above, Theorem 2 follows since we can compute each $\overline{r}(u, v)$ from the effective resistances in $\mathcal{S}$ and can compute $G$ from these resistances by Theorem 1. □

**Gradient and Hessian Computations.** We now show closed form formulas for the gradient and Hessian of Problem 2, using the notation established in Section 3.2.

**Proposition 1.** *Let $\circ$ denote the Hadamard (entrywise) product for matrices. Define the error vector $\Delta(w) \in \mathbb{R}^m$ as having $\Delta(w)_i = \overline{r}(i) - [R(w)]_{i,i}$ for all $i \in \mathcal{S}$ and 0s elsewhere. We have:*

$$\nabla F(w) = 2 \left( R(w) \circ R(w) \right) \Delta(w).$$

*Proof.* We begin by observing that, letting $e_i$ be the $i$th standard basis vector in $\mathbb{R}^m$, for any weight vector $w$, the graph Laplacian corresponding to the weight vector $w + \epsilon e_i$, is $L(w) + \epsilon b_i b_i^T$. The Sherman-Morrison formula for the matrix pseudoinverse yields:

$$(L(w) + \epsilon b_i b_i^T)^+ = L(w)^+ - \epsilon \frac{L(w)^+ b_i b_i^T L(w)^+}{1 + \epsilon b_i^T L(w)^+ b_i},$$

and, hence thinking of $L(w)^+$ as a matrix-valued function of $w$,

$$\frac{\partial L(w)^+}{\partial w_i} = \lim_{\epsilon \to 0} \frac{1}{\epsilon} \left[ L(w)^+ - \epsilon \frac{L(w)^+ b_i b_i^T L(w)^+}{1 + \epsilon b_i^T L(w)^+ b_i} - L(w)^+ \right]$$
$$= -L(w)^+ b_i b_i^T L(w)^+.$$

Let $R_i$ denote the $i^{th}$ column of $R(w) = BL(w)^+ B^T$. By linearity:

$$\frac{\partial R}{\partial w_i} = -B \left( L(w)^+ b_i b_i^T L(w)^+ \right) B^T = -R_i R_i^T.$$

Thus, $$\frac{\partial F}{\partial w_i} = 2 \sum_{j \in \mathcal{S}} \left( \overline{r}(j) - [R(w)]_{j,j} \right) \cdot [R(w)]_{i,j}^2,$$

and so we obtain that the gradient equals $\nabla F(w) = 2 \left( R(w) \circ R(w) \right) \Delta(w)$. □

While gradient descent works well in our experiments, one may also apply second order methods, which require $F(w)$'s Hessian. By similar computations to those in Proposition 1:

**Proposition 2.** *Let $I_\mathcal{S} \in \mathbb{R}^{m \times m}$ be the diagonal matrix with a 1 at each entry corresponding to $i \in \mathcal{S}$ and 0s elsewhere and $\Delta(w)$ be as defined in Proposition 1. The Hessian matrix of $F(w)$ is:*

$$H_F(w) = -4 \left[ R(w) \operatorname{diag}(\Delta(w)) R(w) \right] \circ R(w) + 2(R(w) \circ R(w)) I_\mathcal{S} (R(w) \circ R(w)).$$

## C  Graph Learning via Convex Optimization – Problem 3

In this section we show how to efficiently solve our convex formulation Problem 3, via a semidefinite program. We can express each effective resistance constraint as a positive semidefinite constraint via a Schur complement condition,

$$\chi_{u,v}^T L^+ \chi_{u,v} \leq \overline{r}(u, v) \quad \text{iff} \quad \begin{bmatrix} L & \chi_{u,v} \\ \chi_{u,v}^T & \overline{r}(u, v) \end{bmatrix} \succeq 0 \text{ and } L \succeq 0.$$

Doing so yields the following program:

> **Problem 4** (SDP Form of Problem 3). *Given vertex pairs $\mathcal{S} \subseteq [n] \times [n]$, and target effective resistance $\bar{r}(u, v)$ for every $(u, v) \in \mathcal{S}$,*
>
> $$\underset{L \in \mathcal{L}}{minimize} \operatorname{Tr}(L)$$
>
> $$subject\ to\ L \succeq 0 \text{ and} \qquad\qquad \forall\, (u, v) \in \mathcal{S}, \begin{bmatrix} L & \chi_{u,v} \\ \chi_{u,v}^T & \bar{r}(u, v) \end{bmatrix} \succeq 0$$

We require $L$ to be a valid graph Laplacian (i.e., constrain $L \in \mathcal{L}$) by adding the linear constraint to the above formulation. Specifically,

$$\forall\, i,\ L_{i,i} = -\sum_{j \neq i} L_{i,j}\ \text{ and } \forall\, i \neq j,\ L_{i,j} \leq 0.$$

## D   Omitted Experimental Results

**Dataset Details.** Details of the synthetic and real work networks studied are shown in Table 2 below.

Table 2:  Datasets for experiments. FB denotes "Facebook".

| Name | # nodes | # edges |
|---|---|---|
| GRID (SYNTHETIC) | 64 | 224 |
| K-NN (SYNTHETIC) | 80 | 560 |
| FB SMALL A (EGO NODE ID 698) | 40 | 220 |
| FB SMALL B (EGO NODE ID 3980) | 44 | 138 |
| FB SMALL C (EGO NODE ID 414) | 148 | 1692 |
| FB SMALL D (EGO NODE ID 686) | 168 | 1656 |
| FB MEDIUM A (EGO NODE ID 348) | 224 | 3192 |
| FB MEDIUM B (EGO NODE ID 0) | 324 | 2514 |
| FB LARGE A (EGO NODE ID 3437) | 532 | 4812 |
| FB LARGE B (EGO NODE ID 1912) | 795 | 30023 |

**Performance on K-NN.** In Figure 4 we show the objective function and generalization error (described in Section 4.2) of the least squares formulation of Problem 2 on the synthetic clustered K-NN graph. As with the GRID graph, we see that gradient descent converges for a variety of noise levels and constraint sets. Convergence leads to improved generalization error.

Figure 4:  Objective error and generalization error for Problem 2, as defined in (5) for K-NN.

# E   Extensions to Other Similarity Measures

As discussed, our results generalize to random walk-based node similarities beyond effective resistances, such as personalized PageRank scores. Given localization parameter $\alpha \geq 0$, the personalized PageRank score $p_G^\alpha(u, v)$ is the probability that a lazy random walk on graph $G$ which jumps back to $u$ with probability $\alpha$ in each step is at node $v$ in its stationary distribution [39, 22, 4].

Letting $W = \frac{1}{2}(I + AD^{-1})$ be the lazy random walk matrix, $p_G^\alpha(u, v)$ is the $v^{th}$ entry of the personal PageRank vector:

$$p_G^\alpha(u) = \alpha(I - (1 - \alpha)W)^{-1}e_u. \tag{11}$$

This vector gives the stationary distribution for the random walk on $G$ and thus the personalized PageRank $p_G^\alpha(u, v)$ is its $v^{th}$ entry.

It is not hard to show an analogous result to Theorem 1, that given a full set of exact personalized PageRank scores, full recovery of $G$ is possible. Roughly, if we let $P$ be the matrix with $p_G^\alpha(u)$ as its $u^{th}$ column, we have $P = \alpha(I - (1 - \alpha)W)^{-1}$ and can thus solve for the random walk matrix $W$, and the graph $G$. This gives:

**Theorem 3.** *For any connected graph $G$, given personalized PageRank score $p_G^\alpha(u, v)$ for each $(u, v) \in [n] \times [n]$, there is algorithm returning $G$ (up to a scaling of its edge weights) in $O(n^3)$ time.*

Further, it is possible to formulate a problem analogous to Problem 2 and solve for a graph matching a subset of personalized PageRank measurements as closely as possible. As shown in Figure 5, personalized PageRank often gives a stronger signal of global graph structure than effective resistance.

To create the plot, nodes are sorted by their value in the Laplacian Fielder vector, which corresponds roughly to residence in different clusters. In an extended version of our work, we will provide detailed empirical results with personalized PageRank, and other random walk measures.

Figure 5: Personalized PageRank correlates better than commute times with the cluster structure in the FB SMALL C network (see Table 2). Heatmaps are shown in log scale.