[Reviews · NeurIPS 2018]

Reviewer 1



Updates after reading author response and other reviews Thanks for the clarity on the egonets. I agree that perhaps the social structure can be interesting and this could be a strength rather than weakness. I didn't actually have a concern on practical use cases, as the response states. I just think it would be more interesting to consider noise that appears from the types of situations described, rather than Gaussian (and the authors acknowledged this point). Overall, this was an interesting submission. I am upping my score to 7. -------------------- This paper proposes to learn an unknown graph given a subset of the effective resistances of pairs of nodes. The proposed methods can substantially out-perform random guessing in network reconstruction on a couple synthetic networks and some Facebook egonets. The main idea of the paper is interesting and original, and the proposed methods do well on the datasets considered. However, I think the choice of egonets as real-world datasets is limiting. The fact that all nodes share a common neighbor implies substantial social structure (and possibly substantial graph structure, if the ego is included --- it is not clear if the ego is included in these datasets). I would suggest to instead use a variety of social networks, or if the authors are keen on Facebook, use some of the Facebook100 graphs. Another weakness in the experimental design is the following. The noise terms for the experiments are Gaussian, but I read the motivation for the noise terms to come from using approximation algorithms to deal with computational constraints or from partial measurements (last paragraph of introduction, lines 43--48). The usefulness of the Gaussian error seems to be that the solution to the optimization problem (Problem 2) is the MLE. The experimental results would be much more convincing if the noise was closer to the motivation. For example, the authors could compute approximations to the effective resistance with an approximation algorithm (the simplest case might be to run just use a few steps of an iterative solver for computing the effective resistances). From the presentation of the introduction, I expected more theoretical results. However, the only formal theoretical statements I see in the main text are Theorem 1, which was proved in another paper, and Proposition 1, which is a gradient calculation. Overall, I think this paper would greatly benefit by tightening up all of the language around the problem formulation and including more experiments and description of the actual methodology (e.g., what is the SDP formulation?). All of that being said, I enjoyed the main idea of the paper. The problem itself opens up some interesting avenues for both theoretical and empirical future work, and the numerical experiments do show that the proposed methods can be successful. I think the experiments (namely, choice of real-world datasets and types of noise) and presentation could be updated for an improved camera version. Thus, I lean towards accepting this paper. Fianlly, perhaps I am missing a compelling reason that we should be using egonets as the main example, so please point this out to me if it is the case. Some small issues: 1. In the explanation of privacy attacks (lines 36--42), it would be useful to know some scenarios under which the node similarities would be released, but the link information would not be released. 2. "we show that, when G is a tree, exact recovery is possible for Problem 1.2 when S is a superset of G's edges" Should this be subset of edges? 3. Please include a reference to [43] in the statement of Theorem 1. Right now, it seems like the theorem is a novel contribution of this paper, which, from my reading of the text, it is not. 4. "This approach quickly converges to near global minimums for many networks." I am not sure how I am supposed to draw this conclusion from the experiments. 5. "This modified problem is convex and can be solved via an SDP." Where is the SDP formulation? 6. Why is there no \sigma^2 = 0.1 case for FB Medium A and FB Large A. 7. Typos: Appendixx E (line 186), we learn via share (line 236)

Reviewer 2



The authors investigate the problem of graph reconstruction via random walks when partial information about node similarities is observed. The problem is very interesting from a data privacy perspective. The paper is well-written, clearly organized, and interesting. Having said that, I am not very well versed in the graph reconstruction or privacy literature, and feel that I cannot comment on the significance or originality of the work.

Reviewer 3



The focus of the paper is on the reconstruction of graphs from (noisy) estimates of the effective resistances between nodes. The main contribution is a gradient descent algorithm on the vector of edge weights (Proposition 1). The effectiveness of the approach is illustrated on both synthetic and real data (small Facebook ego networks). The paper is topical and well written. However, the proposed approach is not very original and applies to small graphs only as it requires a memory of order n^2. The motivation for Problem 1.2 (reconstruction from partial data) is unclear as the proposed solution requires a memory of n^2 in any case. Besides, Problem 1.2 is solved only in the specific case of trees. Detailed comments: * The abstract suggests that the effective resistances can be "measured". What do you mean exactly? * The discussion on privacy should be clarified in the introduction. In which practical case would a social network reveal effective resistances but not links?